

# A comparative study of life skills, lifestyle habits and academic performance in health promoting and non-health promoting schools in the Autonomous Community of Aragon, Spain

Beatriz Sánchez-Hernando[1,2], Ángel Gasch-Gallén[2,3], Isabel Antón-Solanas[2,3], Vicente Gea-Caballero[4,5], Raúl Juárez-Vela[4,6], Javier Gállego-Diéguez[7], María Inmaculada Carboneres-Tafaner[8], Emmanuel Echániz-Serrano[3,9], Laura Lasso-Olayo[3] and Ivan Santolalla-Arnedo[6]

[1] Health Center "Amparo Poch", Aragon Health Care System, Zaragoza, Aragón, Spain
[2] Aragón Health Research Institute, Nursing Research Group in Primary Care of Aragon (GIIS094-GENIAPA), Zaragoza, Aragon, Spain
[3] Department of Physiatry and Nursing, University of Zaragoza, Zaragoza, Aragón, Spain
[4] PBM Research Group, Research Institute IdiPaz, Madrid, Madrid, Spain
[5] Faculty of Health Sciences, International University of Valencia, Valencia, Spain
[6] School of Nursing., Research Group in Cares GRUPAC., Universidad de La Rioja, Logroño, La Rioja, Spain
[7] Head of the Information, Transparency and Participation Service, Health Department., Government of Aragon, Zaragoza, Zaragoza, Spain
[8] CEIP Mare de Deu de la Vallivana., Generalitat Valenciana,Conselleria de Educació, Picassent, Valencia, Spain
[9] Research Group Cultural Transferences and International Projection of Aragonese Culture (H27_20D- TRANSFERCULT), Zaragoza, Aragón, Spain

Corresponding author
Raúl Juárez-Vela,
raul.juarez@unirioja.es

## ABSTRACT

**Background:** There is insufficient evidence about the impact of health promoting schools on the student's health and academic performance. The aim of this study was to compare the life skills, lifestyle habits and academic performance of 7th and 8th grade students registered in health promoting and non-health promoting schools in the Autonomous Community of Aragon.

**Methods:** An observational, descriptive, comparative study of the life skills, lifestyle habits and academic performance of 7th and 8th grade students registered in 43 randomly selected health promoting and non-health promoting schools was carried out. We used an adapted tool, which was validated through the expert panel technique, to evaluate the students' life skills (self-efficacy, social skills and affect balance), lifestyle habits (diet, sleep, physical exercise, use of screen and substance abuse) and academic performance.

**Results:** The socioeconomic level of the mothers and fathers of the students who attended non-health promoting schools was significantly higher than that of the parents of the adolescents who attended health promoting schools ($p < 0.001$). The students who attended non-health promoting schools had better self-efficacy, led a healthier life generally and achieved better academic results ($p = 0.03$).

**Conclusions:** As opposed to previous studies, which found a positive association between health promoting schools and life skills, lifestyle habits and academic performance, our findings suggest that the impact of these health programs on the students' health and academic performance is limited. We recommend that the degree of implementation of health promotion programs in health promoting schools is systematically evaluated, and the health promoting school accreditation process and requirements reviewed, in order to ensure that the health of school children and adolescents is safeguarded and promoted in the short, medium and long term.

## INTRODUCTION

In 1986, the World Health Organization (WHO) through the Ottawa Charter for Health Promotion (*World Health Organization, 1996*) defined health promotion as "the process of enabling people to increase control over, and to improve, their health". In the compulsory education context, health promotion is defined as any activity or intervention carried out with the aim of improving or protecting the health of those who work, learn and coexist in a school (*St Leger et al., 2010*). According to the WHO (*World Health Organization, 1996*), a school can be classed as a health promoting school (HPS) when it creates a health fostering environment that promotes individual, family and community responsibility through the development of specific knowledge and skills that enable its members to make healthy choices and adopt and a healthier physical, psychological and social lifestyle, whilst implementing a coherent and participative curriculum. This is underpinned by the idea that beyond just teaching academics, schools should foster students and other members of the school community's health, wellbeing and social development through two main strategies: (1) health promoting recommendations and guidelines, and (2) consideration of previous experiences, scientific and technical knowledge of health and education, including theories, models and educational activities (*World Health Organization, 1996*; *St Leger et al., 2010*; *Aliaga et al., 2016*; *World Health Organization, 2006*; *Brooks, 2014*). To this end, the WHO defined global standards for HPS, the key objectives of which were: to generate scientific evidence, develop a set of standards that can be adapted to different contexts, develop a common framework to monitor and evaluate them, develop a guide to implement them, create a web platform and provide technical support for the application and adaptation of these standards (*World Health Organization & UNESCO, 2021*).

Based on the above, it seems reasonable to assume that students who attend a HPS are more likely to achieve better academic results and health outcomes than those who attend a non-HPS (*St Leger et al., 2010*; *World Health Organization, 2006*). This assumption is based on the idea that an association exists between certain social and emotional skills and competencies and the adoption of a healthier lifestyle; that is, attending a HPS that

provides a favorable physical and socioeconomic environment allows students to achieve a higher level of health and wellbeing that, in turn, optimizes their learning disposition (*Brooks, 2014*). Some previous studies (*Durlak et al., 2011*; *Yuasa et al., 2015*) have discussed the benefits of educational programs that are based on health promotion; some have even demonstrated an association between HPS and improved academic performance. Other studies have analyzed the relationship between HPS and health outcomes, suggesting that students who attend HPS have better health in general and better self-perceived health than students who do not (*Lee et al., 2019*).

Other studies have analyzed the relationship between HPS and lifestyle habits. Specifically, researchers have investigated the impact of HPS on nutrition (*Yuasa et al., 2015*; *Lee et al., 2019*; *Arthur et al., 2011*; *Passmore & Donovan, 2014*), reporting an increase in mid-morning snacks, better dietary habits and nutritional awareness (*Arthur et al., 2011*), enhanced cooking skills and improved knowledge of food growing practices (*Passmore & Donovan, 2014*). An association has also been discovered between HPS and PA (PA) (*Yuasa et al., 2015*; *Lee et al., 2019*; *Arthur et al., 2011*; *Passmore & Donovan, 2014*) in terms of better and more comprehensive choices for exercising, new and innovative strategies to promote walking and cycling to school, and better support of sport clubs (*Arthur et al., 2011*). Further, some authors have suggested that an association exists between HPS and general hygiene practices (*Yuasa et al., 2015*), enhanced social and emotional skills, better behavior, improved school and family atmosphere (*Durlak et al., 2011*; *Yuasa et al., 2015*), better satisfaction with one's life (*Lee et al., 2019*), a better level of mental health (*Lee et al., 2019*) and even higher levels of motivation of the teaching staff (*Yuasa et al., 2015*).

Some researchers (*Bonde et al., 2018*; *Moynihan, Jourdan & Mannix McNamara, 2016*), however, have not been able to demonstrate an improvement in health outcomes and/or academic performance of students who attended a HPS compared to students who attended a non-HPS. Moreover, a Cochrane systematic review (*Langford et al., 2014*) assessing the effectiveness of the WHO HPS framework for improving the students' health, wellbeing, and academic performance concluded that there is limited evidence to suggest that the HPS approach can improve specific areas, as well as fat intake, alcohol and drug use and academic outcomes; due in part to the low to moderate quality of the studies included in the review.

In the Autonomous Community of Aragon there is a range of schools and levels of integration of health promoting strategies in education. Each school includes health-related content in their curricula, but the level of integration and implementation of these approaches varies from one school to the next. The schools which, theoretically, propose a higher degree of integration and implementation of health promoting approaches are those integrated in Aragon's Network of HPS (ANHPS). Similar to the Schools for Health in Europe (*Schools for Health in Europe, 2021*) and the Latin-American Network of HPS (*Organización Panamericana de la Salud, 2018*), ANHPS has established an accreditation system for schools that wish to become accredited as HPS. This accreditation system is based on a range of requirements in relation to the school's management, actions to address the Determinants of Health, health education programs,

relationship between the school and the wider community, and promotion of student life skills. The criteria that are considered mandatory for the accreditation of the centers are: a commitment of at least 3 years; the integration of education and health promotion on an ongoing basis; the creation of a work team, supported by the management; the interaction of the center with the services of the school environment; and the periodic evaluation of the actions carried out. The evaluation is performed by a commission integrated by members of the Health Service and the Education Service of the Government of Aragon (Aliaga et al., 2016).

Given that the quality of the evidence currently available on the effect of HPS on students' academic performance and health outcomes is uneven, it is important to carry out new investigations on large samples in order to support and improve these and similar initiatives to integrate health promotion in compulsory education. Thus, the aim of this investigation was to compare the life skills, lifestyle habits and academic performance of students aged 12–15 registered in a sample of HPS and non-HPS in the Autonomous Community of Aragon.

## MATERIALS AND METHODS

### Design

An observational, descriptive, comparative study of the life skills, lifestyle habits and academic performance of students aged 12–15 attending HPS and non-HPS in the academic year 2018–2019 was carried out.

### Participants and study location

The study population comprised all the 7th and 8th grade middle school students registered at some of the 185 middle schools existing in Aragon in the academic year 2018–2019 (N = 27,184). This information was obtained from the Institute of Statistics of Aragon (INE, 2021).

We excluded schools of special needs education, in order to guarantee the homogeneity of the sample, and grouped rural schools, as students that attend these schools are not within the age range of the study population. We estimated a minimum sample of 379 participants, with a 95% confidence interval and 5% margin of error. Sampling was carried out in two phases. First, we categorized all the middle schools into public or private, urban or rural and HPS or non-HPS; subsequently, we applied a cluster sampling technique using the online application "Research Randomizer" (www.randomizer.org) in order to ensure the representativity and homogeneity of the sample based on the criteria mentioned above. A total of 106 middle schools were selected following this process; of the 106 middle schools selected, 43 accepted to participate in this study. In the academic year 2018–2019, there were a total of 5,132 7th and 8th grade students registered at the 43 middle schools selected. Second, we used universal sampling to select the individual participants. Of the 5,132 7th and 8th grade students, 1,745 gave their consent to participate in this study. As the participants were underage, written consent from the students' mother, father or legal tutor was also obtained before commencing data collection. We excluded students who could not communicate in Spanish.

## Data collection

As this study is one part of a larger project, a doctoral thesis dealing with health promotion in the school environment, data were collected as previously described by *Sánchez-Hernando et al. (2021a)*, *Sánchez-Hernando et al. (2021b)* during the same study period (2021), specifically, the participants completed a self-administered, anonymous questionnaire throughout April 2019. This tool was adapted from four previously validated instruments, namely the questionnaire designed and implemented by the Health Behavior in School-Age Children (HBSC) study (*Moreno et al., 2016*) carried out in Spain in 2014, the general self-efficacy scale (*Espada et al., 2012*), the first subscales of the social skills assessment scale (*Goldstein et al., 1980*), and the affect balance scale (*Godoy-Izquierdo, Martínez & Godoy, 2008*). The adaptation and validation of the self-administered questionnaire was carried out using the expert panel technique (*Masdeu, 2015*). We recruited six experts who took part in two sessions, lasting approximately 90 min each, in December 2017. The participants received information about the confidentiality of the information and the protection of the data, with the parents signing the informed consent. The study was conducted according to the guidelines of the Declaration of Helsinki and approved by the Institutional Review Board of the Ethics Committee of the Autonomous Region of Aragón CEICA 18-216 TA.

Internal consistency or homogeneity was good with a Cronbach's alpha coefficient of 0.8465; the separate values obtained for each of the factors was close to one, which indicated that factor analysis for each of the factors was consistent. The instrument's validity was evaluated using different indicators (NFI = 0.802; RMSEA = 0.067; CFI = 0.891; SRMR = 0.093) and showed good adjustment (*Tur-Porcar et al., 2020*).

The final version of the instrument comprised 87 items divided into four dimensions, namely sociodemographic characteristics, life skills, life style habits and academic performance. In turn, each dimension was subdivided into a variable number of subscales.

The sociodemographic characteristics dimension comprised one single subscale including the following variables: sex, year of study, age, parental educational level (both mother and father), perceived level of health.

The life skills dimension, as previously described by *Sánchez-Hernando et al. (2021b)*, integrated three subscales:

- The self-efficacy subscale (10 items) is measured on a four-point Likert scale identifying the degree of applicability of ten statements ranging from untrue of me (1) to true of me (4). The global score ranges from 10 to 40, with lower values indicating a lower level of self-efficacy.
- The social skills subscale (14 items) is measured on a five-point Likert scale and analyzed the frequency of certain behaviors ranging from never (1) to always (5). The global score ranges from 14 to 70, with the results being classified as follows: low level of social skills (0–17), low-average level of social skills (18–29), average level of social skills (30–40), average-high level of social skills (41–52) and high level of social skills (53–70).
- The affect balance subscale (18 items) is measured on a three-point Likert scale ranging from never or almost never (1) to always or almost always (3), quantifying the frequency

with which the adolescents experienced certain emotions. The subscale comprises nine positive items and nine negative ones. Each positive item is scored from 1 to 3 and each negative one from −1 to −3. The global score is calculated by adding the total of positive and negative points; values <0 indicate a negative affect balance whilst values >0 indicate a positive affect balance. The higher the score, the higher the level of affect balance.

The lifestyle habits dimension, as previously described by *Sánchez-Hernando et al. (2021a)*, comprised five subscales:

- The diet subscale (nine items) comprised the following items: breakfast during the week (yes; no), weekly consumption of fruit and vegetables, chips or salty snacks, sweets, soft drinks or sugary drinks, meat, fish and milk or dairy products (up to once per week; 2 to 6 times a week; at least once a day).
- The sleep subscale (1 item) measured the number of hours of nighttime rest during the week (less than 7 h; from 7 to 9 h; more than 9 h).
- The physical activity subscale (five items) assessed the weekly frequency of PA during leisure time (never to once a month; 1 to 3 times a week; 4 to 7 times a week), number of hours of PA during leisure time a week (1 h or less; 2 to 3 h; more than 4 h), playing or practicing team sports and PA (less than 3 times a month; 1 to 3 times a week), practicing individual PA (less than 3 times a month; 1 to 3 times a week), means of transport to school (walking, cycling, car, bus).
- The use of screens subscale (six items) measured time spent daily playing games, watching TV, videos, and other displays on a screen, and using screens for homework or use of social networks (weekdays and weekends) (less than 2 h; 2 h; more than 2 h).
- The substance use subscale (17 items) comprised the following items: use of tobacco (yes; no), use of alcohol, including wine, mixed alcoholic drinks, liquor shots, and other beverages (never; rarely; daily-monthly), use of other substances including cocaine, hashish or marijuana, ecstasy or pills, amphetamines or speed, non-prescription drugs, LSD, glue or solvents, other drugs (never; at least once), age of tobacco use onset (never; younger than 11 to older than 14 years), age of alcohol use onset (never; before 13; after 13), age of first binge-drinking episode (never; before 13; after 13), age of hashish or marijuana use onset in the form of a joint (never; from less than 11 to more than 14 years).

Finally, the academic performance dimension measured the variable final grade, obtained by calculating the mean score of each of the first and second trimester subjects. It was assessed using the students' average score for the whole academic year; in Spain, the academic score is a number from 0 to 10.

## Data analysis

Data codification, processing and analysis were completed using the statistical software STATA/SE v16.0. (StataCorp. 2020, Galveston, TX, USA). Categorical variables were presented using frequencies and percentages; numerical variables were presented using

mean and standard deviation. To analyze the possible significant differences between the HPS centers and the non-HPS centers, the chi-square test and the Kruskal-Wallis test were used he main estimates were presented with a 95% confidence interval (95%), a margin of error of 5% and a level of statistical significance $p < 0.05$ (statistical significance set at two-tailed $p < 0.05$).

## Ethical considerations

Participant information was dissociated in order to guarantee anonymity and confidentiality according to the Data Protection Regulation (EU) 2016/679 of the European Parliament and the Spanish Organic Law 3/2018. There are no conflicts of interest to report for this study for any of the listed authors, nor did they receive any form of financial compensation. Participation in this study was voluntary; no compensation of any kind was offered to the participants. The participants were informed about the aims and procedures of this investigation and informed consent to participate was given in writing by both the participants and their parents or legal tutors. The study protocol was reviewed and approved by a local research ethics committee (Ref. Num. 18-216 TA). The project was endorsed by the General Directorate of Public Health and the Direction of Innovation, Equity and Participation of the Government of Aragon.

## RESULTS

### Sociodemographic characteristics

A total of 1,745 students completed the questionnaire (response rate 34%). Mean age was 13.03 years (SD 0.82; range 12–16); 54.57% of our respondents were female. No significant differences were found between the HPS and non-HPS students in terms of their sex, age, year of study and health status. We found that 18.33% of the fathers and 10.97% of the mothers of students who attended HPS were educated to primary education level or less, whilst 8.33% of the fathers and 5.53% of the mothers of students who attended non-HPS were educated to the same level. Similarly, we found that the parents of students who attended non-HPS had a higher level of education, with more parents being educated to university level (Table 1).

### Life skills

Students attending non-HPS demonstrated a higher level of self-efficacy than students attending HPS ($p = 0.0019$). No significant differences were found between the students' social skills and affect balance (Table 2).

### Lifestyle habits

No significant differences were found between the groups in terms of the number of hours of nighttime rest during the week. However, we did find significant differences in the rest of the subscales comprising this dimension.

We observed that the students who attended non-HPS consumed more vegetables (23% consumed vegetables once or more times a day) than those attended HPS (17% consumed vegetables ones or more times a day). Similarly, we found significant

**Table 1 Sociodemographic characteristics of the research.**

| Variable | HPS N (%) | No-HPS N (%) | p value |
|---|---|---|---|
| Father's level of education | | | 0.001 |
| Below primary school or no studies | 26 (2.85) | 10 (1.41) | |
| Primary school | 141 (15.48) | 49 (6.92) | |
| Secondary school | 269 (29.53) | 159 (22.46) | |
| Vocational training | 303 (33.26) | 250 (35.31) | |
| Higher education | 172 (18.88) | 240 (33.90) | |
| Mother's level of education | | | 0.001 |
| Below primary school or no studies | 18 (1.92) | 7 (0.97) | |
| Primary school | 85 (9.05) | 33 (4.56) | |
| Secondary school | 253 (26.94) | 139 (19.23) | |
| Vocational training | 320 (34.08) | 201 (27.80) | |
| Higher education | 263 (20.01) | 343 (47.44) | |

**Table 2 Life skills.**

| Variable | HPS Mean ± SD | No-HPS Mean ± SD | p value |
|---|---|---|---|
| Self-efficacy | 30.10 ± 4.53 | 30.75 ± 4.37 | 0.0019 |
| Social skills | 53.86 ± 7.25 | 54.27 ± 7.09 | 0.12 |
| Affect balance | 5.44 ± 5.01 | 5.80 ± 5.03 | 0.13 |

**Note:**
Students attending non-HPS demonstrated a higher level of self-efficacy than students attending HPS.

differences in the frequency of consumption of sugary drinks, with more students attending non-HPS consuming sugary drinks once of less times a day (60%) compared to the students attending HPS (55%).

Students attending non-HPS performed PA more frequently, and spent more hours per week, than those attending HPS. Also, more students attending a non-HPS performed more PA individually and as a group. In terms of transport to school, more students attending a HPS walked or cycled to school, whilst more students attending a non-HPS travelled to school by car or bus.

We found statistical differences between HPS and non-HPS students in terms of use of screens whilst doing homework and use of screen whilst using social media. Also, we observed that 76.86% of HPS students compared to 81.62% of non-HPS students spent two hours or less in front of a screen on weekdays, whilst 57.63% of HPS students compared to 62.63% of non-HPS students spent two hours or less in front of a screen on weekends.

Finally, in terms of substance abuse, we found that fewer students attending non-HPS had consumed hashish or marijuana at least once (0.5%) than students attending HPS (Table 3).

**Table 3 The significance of different variables related to lifestyle habits.**

| Variable | HPS *N* (%) | No-HPS *N* (%) | *p* value |
|---|---|---|---|
| Diet | | | |
| Consumption of vegetables | | | 0.005 |
| Never | 24 (2.48) | 11 (1.45) | |
| Less than once a week | 25 (2.58) | 11 (1.45) | |
| Once a week | 126 (13) | 81 (10.69) | |
| 2–4 times a week | 421 (43.45) | 304 (40.11) | |
| 5–6 times a week | 208 (21.47) | 171 (22.56) | |
| Once a day | 94 (9.7) | 111 (14.64) | |
| More than once a day | 71 (7.33) | 69 (9.1) | |
| Consumption of sugary drinks | | | 0.019 |
| Never | 112 (11.59) | 119 (15.7) | |
| Less than once a week | 151 (15.63) | 145 (19.13) | |
| Once a week | 271 (28.05) | 191 (25.2) | |
| 2–4 times a week | 247 (25.57) | 192 (25.32) | |
| 5–6 times a week | 96 (9.94) | 54 (7.12) | |
| Once a day | 39 (4.04) | 27 (3.56) | |
| More than once a day | 50 (5.18) | 30 (3.96) | |
| Physical activity | | | |
| Weekly frequency of physical activity | | | 0.001 |
| Never | 80 (8.31) | 43 (5.68) | |
| Everyday | 118 (12.25) | 109 (14.4) | |
| 4–6 times a week | 184 (19.11) | 195 (25.76) | |
| 2–3 times a week | 361 (37.49) | 282 (37.25) | |
| Once a week | 145 (15.06) | 108 (14.27) | |
| Once a month | 47 (4.88) | 8 (1.06) | |
| Less than once a month | 28 (2.91) | 12 (1.59) | |
| Number of hours per week of physical activity | | | 0.001 |
| None | | | |
| 30 min | 128 (13.38) | 62 (8.26) | |
| 1 h | 112 (11.70) | 58 (7.72) | |
| 2–3 h | 200 (20.9) | 127 (16.91) | |
| 4–6 h | 326 (34.06) | 271 (36.09) | |
| <6 h | 119 (12.43) | 152 (20.24) | |
| | 72 (7.52) | 81 (10.79) | |
| Physical activity within a group | | | |
| Never | | | 0.004 |
| 2–3 times a month | 246 (26.22) | 148 (20.73) | |
| Once a week | 157 (16.74) | 130 (18.21) | |
| 2–3 times a week | 142 (15.14) | 85 (11.90) | |
| | 393 (41.90) | 351 (49.16) | |

(Continued)
| Table 3 (continued) | | | |
|---|---|---|---|
| Variable | HPS N (%) | No-HPS N (%) | p value |
| Physical activity individually | | | |
| Never | | | 0.001 |
| 2–3 times a month | 248 (26.47) | 148 (19.95) | |
| Once a week | 229 (24.44) | 156 (21.02) | |
| 2–3 times a week | 218 (23.27) | 193 (26.01) | |
| Means of transport to school | | | |
| Walking | | | 0.001 |
| Cycling | 632 (65.49) | 365 (50) | |
| By car | 8 (0.83) | 22 (3.01) | |
| By bus | 149 (15.44) | 153 (20.96) | |
| | 176 (18.24) | 190 (26.03) | |
| Use of screens | | | |
| Homework and social media on weekdays | | | 0.002 |
| None | 70 (7.23) | 71 (9.45) | |
| 30 min | 360 (37.19) | 304 (40.48) | |
| 2 h | 314 (32.44) | 238 (31.69) | |
| 3–5 h | 162 (16.74) | 87 (11.58) | |
| 5–7 h | 51 (5.27) | 30 (3.99) | |
| >7 h | 11 (1.14) | 21 (2.80) | |
| Homework and social media on weekends | | | 0.046 |
| None | | | |
| 30 min | 65 (6.75) | 63 (8.38) | |
| 2 h | 208 (21.60) | 186 (24.73) | |
| 3–5 h | 282 (29.28) | 222 (29.52) | |
| 5–7 h | 217 (22.53) | 174 (23.14) | |
| >7 h | 122 (12.67) | 65 (8.64) | |
| | 69 (7.17) | 42 (5.59) | |
| Substance use | | | |
| Consumption of hashis or marijuana ever | | | 0.020 |
| Never | 961 (98.26) | 749 (98.81) | |
| 1–2 times | 10 (1.02) | 2 (0.26) | |
| 3–5 times | 1 (0.10) | 3 (0.40) | |
| 6–9 times | 0 | 1 (0.13) | |
| 10–19 times | 3 (0.31) | 0 | |
| 20–29 times | 3 (0.31) | 0 | |
| 30 times or more | 0 | 3 (0.40) | |

## Academic performance

With regard to their academic performance, students attending non-HPS achieved higher academic scores that those who attended HPS (Table 4).

**Table 4  Academic Performance of No HPS and HPS.**

|  | HPS | No HPS | *p* value |
|---|---|---|---|
| Academic performance |  |  | 0.03 |
| Mean (SD) | 6,66 (1.33) | 6,81 (1.36) |  |
| Median (IR) | 6,60 (1.93) | 6,75 (2.01) |  |

## DISCUSSION

The aim of this study was to compare the life skills, lifestyle habits and academic performance of 7[th] and 8[th] grade students attending HPS and non-HPS in the region of Aragón (Spain). Based on the evidence available, we expected that students attending HPS would have better life skills and lifestyle habits, and would achieve higher academic scores, than students attending non-HPS (*St Leger et al., 2010*; *World Health Organization, 2006*; *Brooks, 2014*; *World Health Organization & UNESCO, 2021*; *Durlak et al., 2011*; *Yuasa et al., 2015*). Yet, our findings indicate the contrary.

Regarding the participants' sociodemographic characteristics, we observed that the level of education of the students' parents was higher for non-HPS than for HPS. Usually, there is a positive association between level of education and socioeconomic status (*Commision on Social Determinants of Health, 2007*). Thus, it is possible that the families of the students who attended non-HPS had a higher socioeconomic status than those who attended HPS according to the WHO's Conceptual Framework for Action on the Social Determinants of Health (*Commision on Social Determinants of Health, 2007*). Based on this model (*Commision on Social Determinants of Health, 2007*), those mothers and fathers with a higher educational level and social status would be able to provide better opportunities for their children, which may in turn have an impact on their health status and academic performance, despite not attending HPS. It is also likely that secondary schools located in areas that contain populations ranking higher in socioeconomic status have adapted their school policies to the specific characteristics and requirements of the students and their families, and that they have done so outside of the HPS program framework.

Contrary to a previous study by *Lee et al. (2019)*, we found no significant differences between the students' self-perceived health status. In addition, taking into account the different socioeconomic status, it would be expected that students from non-HPS centers would have a higher level of health (*Commision on Social Determinants of Health, 2007*), despite this we did not observe this result either. This may be explained by the fact that adolescents and young people in general are more likely to rate their own health as good or very good (*Moreno et al., 2016*). However, more research is needed to ascertain and analyze those factors that influence adolescents' self-perception of health.

On the one hand, the evidence (*Sánchez-Hernando et al., 2021b*) suggests that there is a positive association between life skills and academic performance in middle school students. Accordingly, it would be expected that students attending HPS would not only display better life skills, but also better academic results. On the other hand, the results of
the study of *Wiederkehr et al. (2015)* affirm that there is a positive relationship between socioeconomic status and skills such as self-efficacy. We observed that the students attending non-HPS had better life skills and achieved higher scores than those attending HPS, although only self-efficacy was statistically significant. Our findings contrast with those reported in previous studies (*Durlak et al., 2011*; *Yuasa et al., 2015*; *Lee et al., 2019*), which confirmed a positive association between life skills and attending an HPS, but they are in line with those that relate self-efficacy to socioeconomic level (*Wiederkehr et al., 2015*).

Regarding the students' lifestyle habits, namely diet, sleep, physical exercise, use of screens and substance abuse, we observed that lifestyle habits do have an impact on academic performance (*Sánchez-Hernando et al., 2021a*). Some authors (*Kristjánsson, Sigfúsdóttir & Allegrante, 2010*; *Barchitta et al., 2019*; *Chacón-Cuberos et al., 2018*; *Burrows et al., 2017*) have reported a positive association between healthy eating and academic performance, others (*Morón et al., 2018*; *Portoles & González, 2015*) have found an association between sleep and academic achievement, and others, such as *Kristjánsson, Sigfúsdóttir & Allegrante (2010)* and *Sánchez Pérez (2015)*, confirmed that those students who exercised more, achieved better academic results. Also, as might be expected, unhealthy lifestyle habits have been linked to worse academic results (*Harlé & Desmurget, 2012*; *Rodericks et al., 2018*; *Navalon & Ruiz-Callado, 2017*). In line with this, we expected students attending HPS to lead a healthier life and also to achieve better academic results than those attending a non-HPS (*Sánchez-Hernando et al., 2021a*). However, as opposed to previous studies (*Yuasa et al., 2015*; *Lee et al., 2019*; *Arthur et al., 2011*; *Passmore & Donovan, 2014*), we found that students attending non-HPS consumed more vegetables and less sugary drinks, and exercised more frequently and for longer periods, both individually and as a group, than students attending HPS. Instead, our findings are in line with the Conceptual Framework for Action on the Social Determinants of Health (*Commision on Social Determinants of Health, 2007*). Specifically, we found that the socioeconomic status of the parents whose children attended non-HPS was higher than those whose children attended HPS. Accordingly, we argue that those parents whose social status was higher were able to offer their children a healthier choice of nutrients, thus generally improving the quality of their diet, and encouraged them to engage in physical activity.

Students attending HPS walked or cycled to school more frequently than those attending non-HPS. There may be an association between the students' choice of means of transport and HPS, as HPS programs do encourage their students to take more active options to travel to and from school. Yet, there may be other factors influencing the students' choice of means of transport. For example, it is possible that the range of options to travel to and from school available to the students attending non-HPS was larger than that available to those attending HPS, and that HPS students simply did not have a choice.

According to the Conceptual Framework for Action on the Social Determinants of Health (*Commision on Social Determinants of Health, 2007*), there is an association between use of screens and sedentary behavior. In line with this, we found that the students who attended non-HPS schools, and who also exercised more, spent less time sitting in

front of a screen. In terms of substance abuse, although the percentage of students who had never smoked hashish and/or cannabis was slightly higher amongst those attending non-HPS, we did not find any significant differences between them.

Finally, in terms of academic performance, students attending non-HPS achieved slightly higher scores than students attending HPS, but there was not a significant difference between them. This finding is in agreement with previous studies (*Durlak et al., 2011*; *Yuasa et al., 2015*). It is not surprising that the non-HPS students achieved slightly better academic results that the HPS students, as both their life skills and lifestyle habits were also better (*Sánchez-Hernando et al., 2021a*; *Sánchez-Hernando et al., 2021b*). Also, it is well known that the higher the socioeconomic level of the parents, the better the children's academic performance (*Commision on Social Determinants of Health, 2007*).

Generally, as opposed to previous studies (*Bonde et al., 2018*; *Moynihan, Jourdan & Mannix McNamara, 2016*), we did not find a significant association between HPS and the study variables. This may be due to different factors. One of them could be that the different schools not accredited like HPS could be carrying out health promotion activities following programs outside the network or in a freeway. According to Jourdan et al. the health promotion approach is required to advance both academic and health outcomes but adaptation of school's policies, structures and systems, human resources, and practices is needed (*Jourdan et al., 2021*). Other, according to Lendrum and Humphrey, HPS programs are rarely implemented as planned due to their complexity (*Lendrum & Humphrey, 2012*), resulting in some HPS implementing their health promotion programs only partially (*Bonde et al., 2018*; *Adamowitsch, Gugglberger & Dür, 2017*). In some cases, schools do not meet each and every requirement to become a certified HPS, but they do undertake different health promoting activities (*Silva et al., 2019*). Also, few studies have evaluated the degree of implementation of health promotion programs in HPS, and their results are inconclusive (*Ramos et al., 2013*). Furthermore, previous studies have identified barriers for the implementation of health programs in the school context, namely the lack of engagement and understanding of the students, their families and the local community (*Ramos et al., 2013*; *Clelland, Cushman & Hawkins, 2013*), problems with the coordination (*Ramos et al., 2013*) and management (*Heesch et al., 2020*) of these programs, unclear roles and responsibilities of the agents involved (*Silva et al., 2019*; *Ramos et al., 2013*; *Mannix-Mcnamara et al., 2012*), and lack of human, material and economic resources (*Silva et al., 2019*; *Ramos et al., 2013*; *Heesch et al., 2020*). Other barriers stem from the low level of implication of the teachers (*Jourdan et al., 2011*), their low self-efficacy to influence the students' behavior (*Clelland, Cushman & Hawkins, 2013*), a difficulty to personally relate to the health promotion program's aims and objectives (*Ramos et al., 2013*) and not considering school health as a priority (*Saito et al., 2015*). Similarly, there are factors that facilitate the implementation of health programs in HPS, including: (1) continuing education and training courses, and support, for teachers implementing health programs (*Moynihan, Jourdan & Mannix McNamara, 2016*; *Silva et al., 2019*; *Ramos et al., 2013*; *Jourdan et al., 2011*); (2) higher engagement and implication of the school community (*Silva et al., 2019*; *Ramos et al., 2013*); (3) higher level of sustainability and impact of interventions (*Passmore & Donovan, 2014*); (4) intersectoral

collaboration and liaison with the local community (*Heesch et al., 2020*; *Tooher et al., 2017*; *Hung et al., 2014*); (5) promotion of a culture of change and active participation (*Heesch et al., 2020*; *Hung et al., 2014*); (6) dissemination of HPS programs amongst schools, organizations and the general population (*Silva et al., 2019*); (7) integrative and holistic approaches designed, managed and implemented systematically (*Ramos et al., 2013*; *Heesch et al., 2020*; *Lee & Cheung, 2017*; *Samdal & Rowling, 2011*).

Some institutions have made attempts to improve the implementation and impact of health promotion programs in HPS (*Arthur et al., 2011*; *Young, St Leger & Blanchard, 2012*; *Centers for Disease Control and Prevention, 2012*), however, little is known about the impact of these initiatives (*Victoria State Government, 2011*). Our findings suggest that the degree of implementation of health promotion programs in HPS, and/or their impact on the students' life skills, lifestyle habits and academic performance, may be insufficient. More research is needed to identify indicators of behavior change in the school context (*Lee et al., 2019*). Further, we argue that HPS accreditation requirements should be based on specific criteria as determined by the WHO or other health authorities. In addition, the process of HPS accreditation should be ongoing and systematic and HPS certificates should be renewed periodically (*Silva et al., 2019*).

As a limitation, it is worth highlighting the low response rate obtained. This could be due to the voluntary nature of the study and the refusal of some families to provide information on sensitive topics. Despite this, the number of participants is still quite high.

## CONCLUSIONS

Significant differences between HPS and non-HPS did not demonstrate the superiority of HPS in terms of life skills, daily habits and academic performance. We recommend that the degree of implication and implementation of health programs in HPS in the region of Aragon are systematically evaluated, and the HPS accreditation requirements reviewed. Health and education commissions should safeguard and protect school children and adolescents' health in the short, medium and long term. More longitudinal and intervention design studies are needed to throw light on the impact of health promotion initiatives in the school context. Future research should also consider socioeconomic status in the evaluation of health promotion programs.

## ACKNOWLEDGEMENTS

The authors express acknowledgement the Government of Aragon and the Public Health Direction.

### Funding
The authors received no funding for this work.

### Competing Interests
The authors declare that they have no competing interests.

## Author Contributions

- Beatriz Sánchez-Hernando conceived and designed the experiments, prepared figures and/or tables, and approved the final draft.
- Ángel Gasch-Gallén performed the experiments, authored or reviewed drafts of the paper, and approved the final draft.
- Isabel Antón-Solanas performed the experiments, authored or reviewed drafts of the paper, and approved the final draft.
- Vicente Gea-Caballero conceived and designed the experiments, prepared figures and/or tables, and approved the final draft.
- Raúl Juárez-Vela conceived and designed the experiments, prepared figures and/or tables, and approved the final draft.
- Javier Gállego-Diéguez analyzed the data, prepared figures and/or tables, and approved the final draft.
- María Inmaculada Carboneres-Tafaner analyzed the data, prepared figures and/or tables, and approved the final draft.
- Emmanuel Echániz-Serrano analyzed the data, authored or reviewed drafts of the paper, and approved the final draft.
- Laura Lasso-Olayo performed the experiments, authored or reviewed drafts of the paper, and approved the final draft.
- Ivan Santolalla-Arnedo performed the experiments, prepared figures and/or tables, and approved the final draft.

## Ethics

The following information was supplied relating to ethical approvals (*i.e.*, approving body and any reference numbers):

Ethics Committee of the Autonomous Region of Aragón approved the study (CEICA-18-216 TA).

## Data Availability

The raw data is available in the Supplemental File.

## Supplemental Information

Supplemental information for this article can be found online at http://dx.doi.org/10.7717/peerj.13041#supplemental-information.

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
