# Peer review of "A comparative study of life skills, lifestyle habits and academic performance in health promoting and non-health promoting schools in the Autonomous Community of Aragon, Spain"

_PeerJ, doi:10.7717/peerj.13041_

## Round 0.1 · original submission · Major Revisions

The reviewers are generally positive about your paper, but have a number of good suggestions for improving it.

Reviewer 1 ·

Basic reporting

Thank you for this interesting study examining how the life skills, lifestyle habits, and academic performance of 7th and 8th grade students in health promoting schools and non-health promoting schools compare.

The manuscript provides good background for the present study in the introduction. However, the section outlining the definition of a HPS (lines 60-69) should be made clearer. I recommend reworking this section and including the six features identified by the WHO as key to HPS (World Health Organization, & UNESCO, 2018, Global Standards for Health Promoting Schools).

I recommend the authors rephrase their summary of the Cochrane review of HPS to avoid potential misinterpretation (lines 93-97). The Cochrane review did find some evidence that the HPS framework can improve certain outcomes, namely physical activity, fitness levels, fruit and vegetable consumption, cigarette use, and being bullied. In other areas, the review found no evidence of effectiveness (fat intake, alcohol and drug use, mental health, violence, and bullying others), or not enough data to draw conclusions. Thus in some specific areas, there was evidence to suggest HPS can make improvements. Overall, the Cochrane review could be given more prominence in the literature review in comparison to the other studies cited, considering its rigour.

The authors should review the formatting standards for PeerJ publications and ensure the manuscript meets these (e.g. font, left-justified text). The test statistics, degrees of freedom, confidence intervals and effect sizes should be included in the manuscript, as per the reporting guidelines.

The report is clear and well-written. There is a spelling error in line 268 and in the descriptor for Table 3. The grammar and capitalization of the table descriptors in general should be reviewed.

Experimental design

The research question is well-defined. However, the observational nature of the study limits how much it can add to the existing research into HPS. As the authors conclude at the end of the manuscript, the field requires longitudinal and intervention research designs in order to further our understanding of the potential of HPS to improve children's outcomes. If the study aims to replicate previous results, this should be stated clearly.

In the methods section, there appears to be an error in the description of the study population (lines 123-124) which states all the students came from a single school. The manuscript later says students from 43 schools participated in the study (line 134).

I commend the detail that has been given in the description of the instruments used in data collection and the rationale behind these choices.

The data analysis section (lines 215-221) needs to be expanded upon further in order for the analysis to be reproducible. The description indicates that the only analysis performed was a chi-square test of association between categorical variables and academic performance (lines 218-220). This does not align with the main research question outlined in the introduction.

Furthermore, considering the ordinal nature of much of the data collected, the use of the Mann-Whitney U test would be more appropriate than chi-square, which is well-suited to categorical variables (McCrum-Gardner, 2008, Which is the correct statistical test to use? British Journal of Oral and Maxillofacial Surgery, 46(1), 38-41). The rationale behind the authors' data analysis choices should be made clear.

Validity of the findings

The discussion (lines 281-292) of the potential impact of the differing socioeconomic status (SES) of the students at different schools is valuable for the interpretation of the results. This awareness should also inform the discussion of the potential reasons that no significant association between HPS and the study variables was found. Specifically, it is possible that the students at HPS schools had better health than would be expected considering their SES. That no difference in self-rated health was found between the two groups of students from differing SES is worthy of note. Likewise, although non-HPS students with higher SES scored significantly higher on self-efficacy, there was no difference on the social skills or affect balance subscales.

There is an extended discussion of implementation and the challenges associated with this (lines 343-384). Given the authors' conclusions regarding the importance of implementation and the specific accreditation requirements, they should include discussion of the ANHPS' accreditation requirements and process and how these compare to best practice, such as the WHO's global standards and indicators (World Health Organization, 2021, Making every school a health-promoting school: global standards and indicators for health-promoting schools and systems).

Furthermore, since this study did not examine implementation, I recommend cutting down this section and focusing instead on the potential impact of SES. The authors should consider the literature on whether a difference in outcomes such as social skills and affect balance would be expected between students whose SES differed to the degree found in the study. Including effect sizes in the results would also enhance the interpretation of the differences that were/weren't found. Further discussion could consider the findings of past HPS studies that did and did not control for SES.

I suggest the recommendations for future research in lines 390-391 should include research that controls for differences in SES.

Additional comments

The aforementioned comments should also be considered in relation to the abstract. In addition, the abstract should be clear that students from non-HPS schools only had better life skills when considering the area of self-efficacy; the other two sub-scales were not significant.

Overall, this is a detailed, well-written study that has the potential to add to the literature on HPS. The unexpected findings are an important contribution to the field. Careful consideration of the methods chosen for data analysis and further scrutiny of the potential reasons for these findings should be undertaken before acceptance.

Reviewer 2 ·

Basic reporting

The paper is clearly written in professional English. The objective of the paper is relevant in the international context, looking both at health and educational outcomes concerning the impact of the HPS approach on students. One of the key findings in the study is that non-HPS schools perform better than HPS schools. Although the paper is well referenced throughout, I recommend the authors to include the recent viewpoint paper in the Lancet CAH, January 2021 by Jourdan et al: Supporting every school to become a foundation for healthy lives. A keypoint is that the study does not look into the level of implementation of the HPS approach in the participating schools. The paper would benefit with a more specific description , specifically concerning line 98-109 how health promotion strategies are implemented in the schools that take part in the investigation. Some more specification of the requirements of the Aragon accrediation system would also be useful for the reader. The main issue is that a school that is not part of the Aragon HPS network can still implement HPS related activities. This will give a stronger argument to one of the conclusions which is that the process of implementation is crucial, from line 341.

Experimental design

The design of the study is well developed. The research question is well developed and relevant. If clearly fills a research gap on the system of accreditation of HPS. Concerning data collection the authors mention in line 142 that it is part of a larger project. This could be more explained.

Validity of the findings

Concerning the response rates ( line 235) 34% seems rather low. Maybe the authors could make a comment on this in the discussion.
Findings are solid and robust and clearly formulated.

Conclusions are clear and give an answer to the research question.

I recommend that the authors include the new WHO global standards and indicators for HPS (2021) and/or the European standards and indicators in their discussion of the results (line 379)

Additional comments

In line 289 a word seems missing, please check.

---

## Round 0.2 · Minor Revisions

Thank you for the revised version. One of the reviewer suggests some further minor revisions, which I recommend you consider.

Reviewer 2 ·

Basic reporting

The 2nd version has improved, and my comments are all included.
My main comment is that there seems to be no chapter on Conclusions. What is written under Conclusions are Recommendations in my opinion
The discussion chapter is now rather long. It may be useful to group the different issues and arguments better. The main argument is that being an HPS is not a predictor for positive health and learning outcomes. It is the level of implementation that matters.
The authors could be more consistent in the use of the term school, sometimes center, sometimes institutions is being used. This is a bit confusing.

line 43: In the abstract it says in Conclusions: As opposed to previous studies, etc. I recommend to write: ‘As opposed to some previous studies, …
Line 67: this reference 6 should be replaced based by the report on the HPS global standards 2021 (WHO & UNESCO (2021). Making every school a health-promoting school: global standards and indicators for health-promoting schools and systems. Geneva: World Health Organization and the United Nations Educational, Scientific and Cultural Organization; 2021).
Line 76: Previous studies..: add: Some previous studies..

Experimental design

no new comments

Validity of the findings

no new comments

---

## Round 0.3 · accepted · Accept

Thank you for the changes, which have improved the paper.